# GeoSES: A socioeconomic index for health and social research in Brazil

Ligia Vizeu Barrozo[1,2,3], Michel Fornaciali[4], Carmen Diva Saldiva de André[5], Guilherme Augusto Zimeo Morais[4], Giselle Mansur[1,2], William Cabral-Miranda[1,3], Marina Jorge de Miranda[6], João Ricardo Sato[4,7], Edson Amaro Júnior[4]*

1 Departamento de Geografia, Faculdade de Filosofia, Letras e Ciências Humanas, Universidade de São Paulo, São Paulo, SP, Brazil, 2 Programa de Apoio ao Desenvolvimento Institucional do SUS (PROADI-SUS), São Paulo, Brazil, 3 Instituto de Estudos Avançados, Universidade de São Paulo, São Paulo, SP, Brazil, 4 Hospital Israelita Albert Einstein—Big Data Analytics, Morumbi, São Paulo, SP, Brazil, 5 Departamento de Estatística, Instituto de Matemática e Estatística, Universidade de São Paulo, São Paulo, SP, Brazil, 6 Departamento de Análise em Saúde e Vigilância de Doenças não Transmissíveis (DASNT), Secretaria de Vigilância em Saúde (SVS), Ministério da Saúde (MS), Brasília, DF, Brazil, 7 Centro de Matemática, Computação e Cognição, Universidade Federal do ABC, São Paulo, SP, Brazil

* Edson.Junior@einstein.br

**Data Availability Statement:** All relevant data are within the manuscript and its Supporting Information files.

## Abstract

The individual's socioeconomic conditions are the most relevant to predict the quality of someone's health. However, such information is not usually found in medical records, making studies in the area difficult. Therefore, it is common to use composite indices that characterize a region socioeconomically, such as the Human Development Index (HDI). The main advantage of the HDI is its understanding and adoption on a global scale. However, its applicability is limited for health studies since its longevity dimension presents mathematical redundancy in regression models. Here we introduce the GeoSES, a composite index that summarizes the main dimensions of the Brazilian socioeconomic context for research purposes. We created the index from the 2010 Brazilian Census, whose variables selection was guided by theoretical references for health studies. The proposed index incorporates seven socioeconomic dimensions: education, mobility, poverty, wealth, income, segregation, and deprivation of resources and services. We developed the GeoSES using Principal Component Analysis and evaluated its construct, content, and applicability. GeoSES is defined at three scales: national (GeoSES-BR), Federative Unit (GeoSES-FU), and intra-municipal (GeoSES-IM). GeoSES-BR dimensions showed a good association with HDI-M (correlation above 0.85). The model with the poverty dimension best explained the relative risk of avoidable cause mortality in Brazil. In the intra-municipal scale, the model with GeoSES-IM was the one that best explained the relative risk of mortality from circulatory system diseases. By applying spatial regressions, we demonstrated that GeoSES shows significant explanatory potential in the studied scales, being a compelling complement for future researches in public health.

**Funding:** The work was supported by Ministério da Saúde (PROADI-SUS) 25000.028646/2018-10, the Fundação de Amparo à Pesquisa do Estado de São Paulo – FAPESP (Grant nº 13/21728-2 to LVB and CDSA), and the Conselho Nacional de Desenvolvimento Científico e Tecnológico (Grant nº 301550/2017-4 to LVB).

**Competing interests:** The authors have declared that no competing interests exist.

## Introduction

The ZIP code paradigm says that the place where a person lives is a more critical health predictor than their genetic code [1]. In fact, at the individual level, the relationship between the socioeconomic status (SES) and the prevalence of chronic diseases presents an evident inverse linear gradient [2]. In other words, as SES improves, prevalence diminishes correspondingly. This gradient is reliable and consistent in the relationship found with cardiovascular diseases, type 2 diabetes, metabolic syndrome, arthritis, chronic respiratory tuberculosis, and adverse birth outcomes, as well as violent and accidental deaths [2]. However, individual conditions are not enough to fully explain spatial variation in disease rates and the relationship between social inequalities and health. Area-based studies show that the socioeconomic conditions of places also affect people's health [3–5]. Thus, understanding which characteristics of the socioeconomic environment most explain health conditions is a pressing issue. Such understanding can contribute to the implementation of intersectoral public policies that would be more efficient to improve the health of the population and to reduce inequalities. As individual measures of socioeconomic indicators are rarely available in medical records [6], it is common to use a single geographical variable that summarizes living conditions (e.g., income, education, *per capita* Gross Domestic Product). Although this approach helps to understand how an aspect of the socioeconomic context is related to health, the interpretation of the findings is limited, as the socioeconomic context is multidimensional, involving aspects such as employment, income, education, housing, segregation, mobility, among others [7].

Furthermore, by including more than one socioeconomic variable in a regression model—the most used statistical analysis in ecological design studies—we can violate the underlying assumptions of this analysis due to the effects of collinearity [8]. The use of composite indexes aims to overcome such problems, adding explanatory power to the socioeconomic context of the places. Indexes can be especially useful if they allow us to evaluate how a particular dimension can influence health [9].

In Brazil, studies on health with an ecological approach often use a single variable as a proxy for the socioeconomic context—generally monthly household income or other variables from public statistics. The Human Development Index calculated by the municipality level (HDI-M) [10], is the most often used on the national scale. In addition to presenting construct validity and reliability, it allows for international comparability. However, this index expresses how much the development process guaranteed access to education and culture, conditions of enjoying a long and healthy life, and having an adequate standard of living for the population [11]. Although in practice it has helped to identify development inequalities among Brazilian municipalities, there is mathematical redundancy when it is applied to explain health outcomes by regression models. The HDI longevity component measures the life expectancy at birth, comprising health conditions and risks to morbidity and mortality. Therefore, there is a lack of a synthetic index of socioeconomic conditions prepared from theoretical references for health studies in Brazil.

Considering this gap, here we introduce a tool intended to estimate the burden of the problems attributable to the different social dimensions in health and social research. GeoSES (Geographic Index of the Socioeconomic Context for Health and Social Studies) synthesizes the most relevant socioeconomic dimensions to contextualize health for research purposes, to evaluate and monitor inequalities, and to develop resource and service allocation strategies. We developed the new index for use at three aggregation scales: national (GeoSES-BR), Federative Unit (GeoSES-FU) and intra-municipal (GeoSES-IM, for the 140 municipalities with three or more census sample areas). Without loss of generality, here we present the application on national and intra-municipal scales.

## Methods

### Geographic units

The enumeration area is the smallest geographic entity for which the Brazilian Demographic Census tabulates decennial data, including all household units. It is similar to the Census Block Group in the US Census. Enumeration areas are grouped to form a valid sample area, for which statistical procedures guarantee representation of the whole population. The questionnaires applied to the sample households include the universal basic questionnaire, in addition to others of a more detailed investigation about the characteristics of the household and its residents [12]. Thus, we created the synthetic index by selecting the variables contained in the questionnaire applied to the sample during the 2010 Population Census, comprising about 11% (6,400,000 units) of the Brazilian households.

### Socioeconomic variables and dimensions of context

We based the choice of the variables for the index construction on theoretical studies on health [7,13], keeping seven dimensions of the socioeconomic context. The dimensions (and the number of initial variables in each of them) are education [7], poverty [5], wealth [3], income [1], segregation [5], mobility [6] and, deprivation to resources and services [14]. We list all variables and their meaning on S1 Appendix.

The dimension **income** may influence the etiology of several health outcomes, in part, through mechanisms that involve the acquisition of material resources. **Education** can reflect non-economic features as general and health-related knowledge, the capacity of problem-solving, prestige, influence, social network and access to technological innovation [15] that can bring advantages to the individual's health [9]. **Poverty** refers to absolute poverty, directly linked to the minimum capacity of survival and access to material resources. **Wealth**, on its turn, is different from income, since it is a proxy for all long-life economic resources [9]. Our definition of **material deprivation and access to public services** bases on Townsend's concept of material deprivation [16] that refers to disadvantages concerning other people in the same society to which one belongs. We intend to measure how much material resources and conveniences that are part of modern life (such as adequate housing, car ownership, refrigerator, computer, among others) a person has, also evaluating their access to services, including sanitation, electricity, and internet. **Mobility** can affect a person's health in many ways. One of them concerns the time spent commuting from home to work, which can cause stress on many levels and compromise the time available for study or leisure. In addition, longer commuting time may expose people to higher doses of air pollution in large cities [17]. Finally, residential **segregation** is a broad concept that refers to housing separated from different population groups in different parts of a city [18]. Segregation affects health by intensifying psychosocial effects involving insecurity, anxiety, social isolation, socially dangerous environments, bullying, and depression [14,19,20]. In our analysis, we considered two aspects of segregation: education and income, including levels of income stratified by ethnic groups, which better describe socioeconomic differences in Brazil. We used the Index of Concentration at the Extremes (ICE) to measure residential segregation [14]. Consider the income segregation as an example: the formula uses the number of people who earns more than the 80% percentile, minus the number who earns less than the 20% percentile, divided by the total number of people who have income. The ICE varies from -1 (most deprived) to 1 (most privileged). A negative ICE means that the area presents more people in the condition of deprivation than in the higher extreme; and a positive value, the opposite. A value of zero indicates that the area is not dominated by extreme concentrations of either of the two groups. The calculations of the other segregation types follow the same formula.

## Data collection and processing

The data used to generate the index comes from the 2010 Census, made available by the Brazilian Institute of Geography and Statistics (IBGE), organized by Federative Unit (FU), including questionnaires about "People" (information at the *individual* level, e.g., income, level of education, time spent commuting to work, etc.) and their "Households" (e.g., the existence of material goods, access to resources, etc.). The first step is to gather the original data according to theoretical references [7,13] and data availability related to the interest of the study. Original data contains quantitative information on the variables of interest, for example, "the number of people with higher education in a municipality".

The second step is to process the original data generating the variables of interest of the methodology, which means translating the original information into percentage values. In our example, when quantifying people in a municipality who have completed higher education, the processing result is a value that represents the percentage (from 0 to 100) of that population that has completed higher education. Exceptions apply to variables of direct significance, such as income, in which the average values were considered. During the second step our calculations always considered the weight of the sample area, since the questionnaires represent samples from the regions from which they were collected.

The third step refers to grouping the generated variables into single CSV files considering the scope of the index to be generated, that is, for a municipal scope the information was made available by sample areas; for a statewide coverage, information was made available on a consolidated basis for each municipality of that FU.

At the end of this process, we have the information in the expected format for index calculation, separated by FU and type (sample area or municipality). Also, each file type (sample area or municipality) contains the generated variables for "People" and "Household", consolidating the desired information in a single source.

## GeoSES definition

We developed the index by successively applying the principal component analysis (PCA) technique [21], based on the methodology of Lalloué et al. [22], with a modification in Step 1, considering the correlation matrix of the original variables. The principal components are uncorrelated linear combinations of the original variables and are constructed so that the first component has the maximum variance, the second has maximum variance and is uncorrelated with the first one and so on. The maximum number of components is equal to the number of variables in the study, but in general, it is possible to explain practically all the variability of the data with a smaller number of components. We used the data described above, selected according to the area of interest under analysis, which can be: national (the whole country, using consolidated data from all municipalities), FU (an indicator of each state, using data from its municipalities) or intra-municipal (for a specific municipality using sample area data). The steps for index generation are the same regardless of the area of interest. A project in the municipality of São Paulo was initially developed [23] with the later application on a national scale.

Alternatively, a method of factor analysis could also be used. However, we would have to assume an a priori model. Furthermore, it is well known that in the Principal Components Solution of the Factor Model, which does not suppose any multivariate distribution for the data, the factor loadings are the scaled coefficients of the principal components. Thus, the results obtained by PCA and factor analysis would be, in this case, equivalents.

**Preprocessing.** The calculation of the index starts by reading the data described in the previous section. Then, we add a constant equal to 10 to all read values. The purpose of this

sum is to avoid the instability of the method during matrix inversion. The choice of value 10 is random, but any constant number would have the same effect, without interfering in the final result. That is, since we added the same value to all data, its relative differences remain the same.

**Steps.** 1) The objective in this first step was to generate aggregate indices within each of the dimensions, and a PCA was performed in each of them. The number of components selected was such that the percentage of total variance explained was greater than or equal to 75%. For ease of interpretation, we considered the variables with the highest coefficient in each component.

2) Considering the variables selected in step 1, we applied another PCA, and we considered its first principal component. The objective here was to bring all dimensions together into an overall component.

3) To eliminate variables that have little contribution to the index obtained in step 2, only those whose absolute coefficient values were below the average of the coefficients were eliminated, and we applied the PCA method to the remaining variables. The resulting first component defines the GeoSES (Socioeconomic Index of Geographic Context for Health and Social Studies).

4) The GeoSES values (scores) were then calculated.

5) The scores were standardized for the interval -1 to 1.

Fig 1 compares the development processes of HDI and GeoSES.

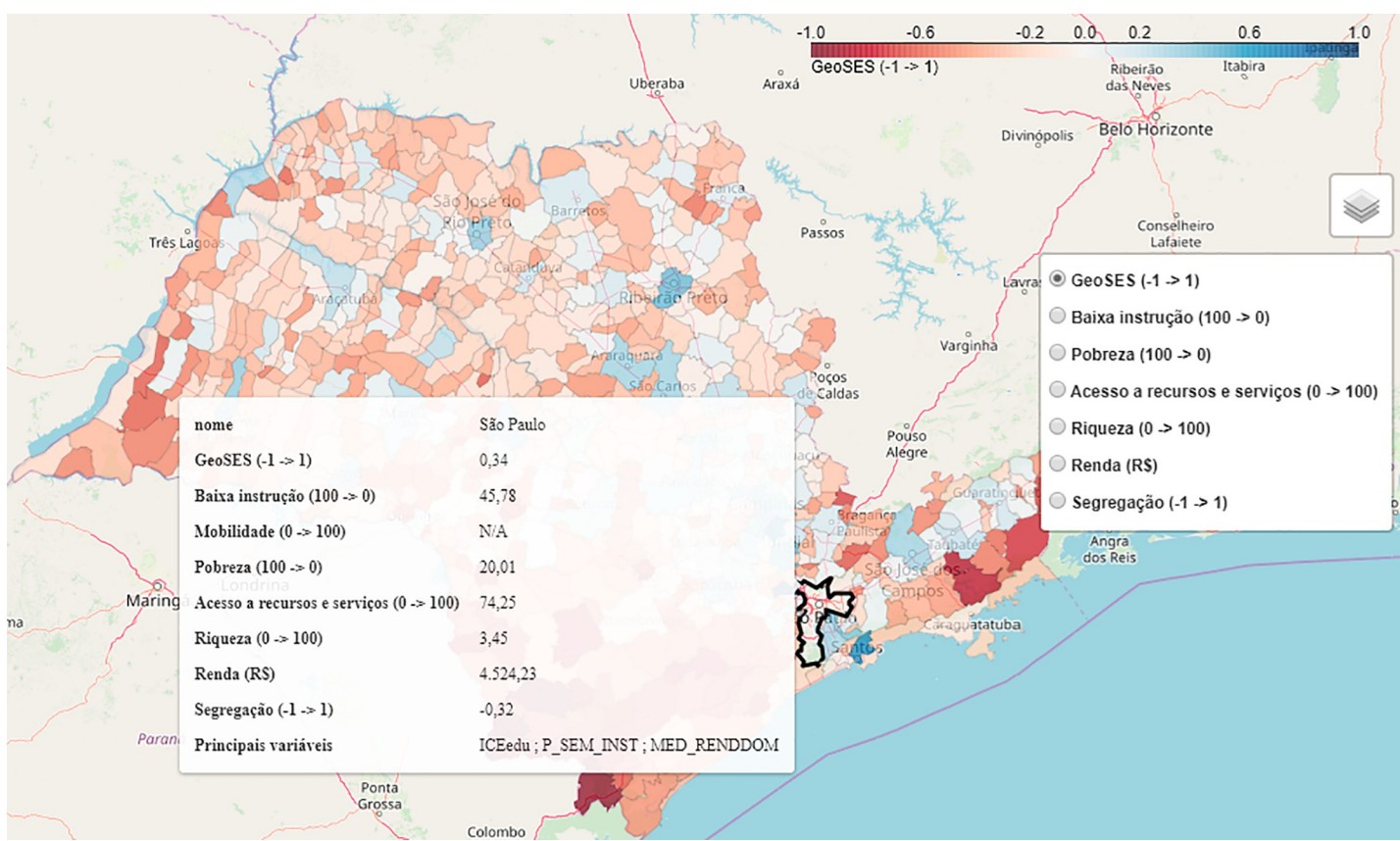

**Fig 1. The development processes of HDI and GeoSES.** The figure shows the dimensions each index employs and how the variables are combined to produce a single measure. Dimensions with the same color are present in both HDI and GeoSES. Image is best seen in color.

The index obtained in Step 2 could have been considered the final one. However, for its calculation, we would need to know the values of variables that do not have an important contribution. In Step 3, these variables were eliminated, and the coefficients of the remainder were calculated again, following the procedure presented in Lalloué et al. [22].

**Interpretation.** Extreme GeoSES values mean the worst (-1) and the best (1) socioeconomic contexts in the analyzed scale. In other words, to have a GeoSES of 1 at the national level, the municipality must have the best *relative* socioeconomic context of the country. So, if in 2010 a municipality had a GeoSES index of 0.2 and in the next Census the index is 0.3, there would have been a *relative* improvement in the socioeconomic context because even with possible changes in the most discriminating variables, this municipality is closest to the best municipality than it was in 2010.

Just as we can decompose the HDI into Longevity, Income, and Education, we developed an approach of expressing the contribution of each GeoSES dimension to better interpretability of the results. We consider that the most significant variable in each dimension expresses its contribution to the index. So, if in the FU analysis of São Paulo, the most relevant variable in the Education dimension is the "percentage of people without education", we use the values of such variable to quantify the development of Education. Thus, considering that for the municipality of São Paulo this value is 45.78, the municipal manager can compare this value with that of other municipalities in the state.

If there is more than one variable in the same dimension, we present only the one most correlated with GeoSES. We understand that by selecting only the most correlated variable, we are using the most representative measure of that dimension. Such a procedure avoids complex processing; also, considering more than one variable per dimension could eventually lead to uninterpretable results.

It is substantial to note, however, that the interpretation of GeoSES as well as of each dimension must consider *comparisons* with one municipality to another. For example, the single value of a GeoSES = 0.2 does not have a meaning *per se*, but the relative differences between the GeoSES of distinct municipalities do, in terms of socioeconomic development.

To facilitate the use and dissemination of results, all indexes and their associated information are available on interactive maps in HTML format. Each municipality or state has a file with its name in which it is possible to observe the geographical distribution of the data interactively. There is a layer on the map for each dimension used in the analysis, plus the prime layer that illustrates the spatial distribution of the index itself. We present an example in Fig 2.

The interactive maps and CSV files, with tabulated values by region of interest, will be made available by the Brazilian Ministry of Health in a place to be defined until the time of publication of this article.

**GeoSES extensive creation.** Once defined the methodology described above and validated for the municipality of São Paulo, using statistical software, the process was completely automated via computer systems. We developed a computer system based on Python language, allowing the scalability of the index generation for the entire national territory, ensuring agility and consistency of results.

For computational development, we adopted a modularized approach for dimensions' parameterization. We facilitate the adaptability of the computer program to data from other Censuses (both past and future), by generalizing how the dimensions and their variables are defined and used.

## Results

The GeoSES evaluation comprised its content, its construct, and its applicability to health studies. Content validation verifies the relevance and representativeness of the dimensions that

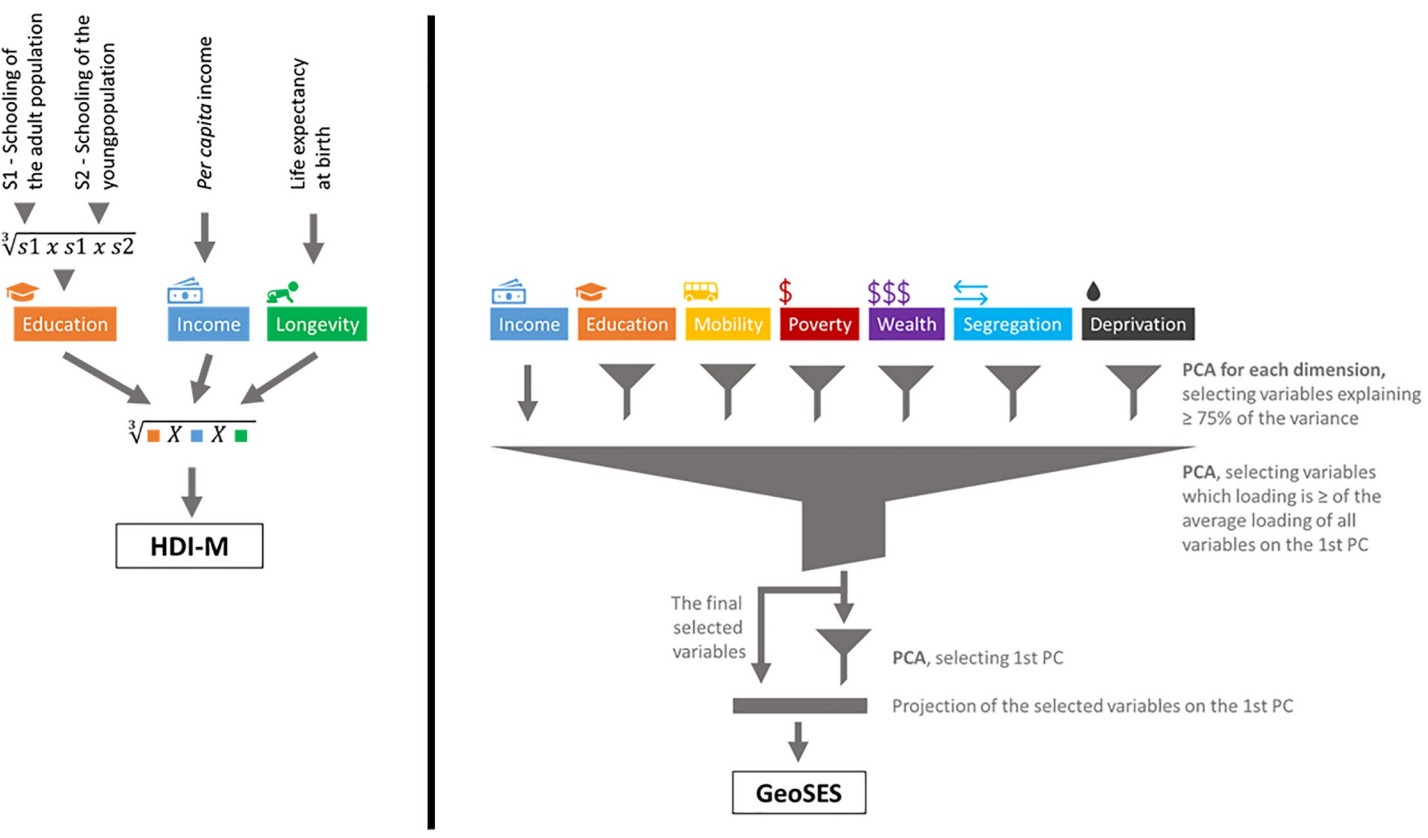

**Fig 2. The GeoSES-SP interactive map.** The figure shows the geographical distribution of GeoSES values, considering the state of São Paulo. In highlight, the city of São Paulo/SP presents its index and the values of its dimensions. In this analysis, we note that the dimension "mobility" is not activated; that is, it is not significant to characterize the socioeconomic differences of the state. Besides the prime layer (the index), we can also plot all other significant dimensions of the analysis.

make up GeoSES to describe the measured phenomenon [24]. Due to the PCA mechanism, the selected variables are naturally representative of the dimensions to which they belong, and the dimensions that make up GeoSES are relevant to contextualize the socioeconomic phenomena due to the theoretical frameworks. Besides, we calculated Cronbach's alpha coefficient for the generated indices, obtaining the values of 0.93, 0.89, and 0.97 for GeoSES-BR, Geo-SES-FU, and GeoSES-IM, respectively. Remember that the closer to 1, the more homogeneous are the index variables.

## GeoSES construct evaluation

The construct validation verifies, from the theoretical point of view, whether the new index is associated with the supposed measured concept. We evaluated GeoSES by comparing it with the HDI-M–a widely accepted and applied indicator for the same level of aggregation (in this case, for municipalities). We verified the association between GeoSES and HDI-M qualitatively and quantitatively.

In qualitative terms, the average of GeoSES-BR among the municipalities of Brazil was -0.40. Melgaço, in the FU of Pará, received the worst rating (-1) and Santana de Parnaíba, in the FU of São Paulo, the best (+1). According to the HDI-M, the average HDI of the Brazilian municipalities is 0.659. Melgaço (0.418) also occupies the worst position, but São Caetano do Sul, in the FU of São Paulo, the best (0.862). Such comparison reinforces the similarity between

**Table 1. Correlation matrix between the indices and their dimensions in the national scale.**

| | *GeoSES* | *education* | *poverty* | *deprivation* | *wealth* | *income* | *segregation* | *HDIM* | *HDI-educ* | *HDI-long* | *HDI-inc* |
|---|---|---|---|---|---|---|---|---|---|---|---|
| **GeoSES** | 1 | | | | | | | | | | |
| **education** | -0.86 | 1 | | | | | | | | | |
| **poverty** | -0.96 | 0.81 | 1 | | | | | | | | |
| **deprivation** | 0.93 | -0.74 | -0.90 | 1 | | | | | | | |
| **wealth** | 0.57 | -0.61 | -0.43 | 0.41 | 1 | | | | | | |
| **income** | 0.93 | -0.82 | -0.88 | 0.83 | 0.60 | 1 | | | | | |
| **segregation** | 0.82 | -0.54 | -0.77 | 0.80 | 0.27 | 0.67 | 1 | | | | |
| **HDIM** | 0.94 | -0.93 | -0.93 | 0.86 | 0.53 | 0.89 | 0.70 | 1 | | | |
| **HDIM_educ** | 0.85 | -0.95 | -0.82 | 0.76 | 0.51 | 0.78 | 0.60 | 0.95 | 1 | | |
| **HDIM_long** | 0.82 | -0.71 | -0.83 | 0.78 | 0.41 | 0.76 | 0.64 | 0.85 | 0.70 | 1 | |
| **HDIM_inc** | 0.95 | -0.84 | -0.96 | 0.87 | 0.53 | 0.94 | 0.73 | 0.95 | 0.82 | 0.83 | 1 |

the indices, but highlights that differences may arise, which potentially better explain the socio-economic conditions of the Brazilian regions.

In quantitative terms, Table 1 shows the correlations between "GeoSES vs. HDI-M" and its "Dimensions vs. Components". GeoSES dimensions showed a good correlation with the HDI-M, above 0.85, except for wealth and segregation. Observe that in the national analysis, "education" and "poverty" appear negatively correlated to GeoSES because education is measured in terms of the population without instruction. That is, in the national context, the number of people *without* instruction is more relevant than people *with* instruction for describing socioeconomical differences.

## GeoSES applicability assessment in health

As an additional evaluation, we validated GeoSES' explanatory potential for health outcomes at two aggregation scales: national (GeoSES-BR), and intra-municipal (GeoSES-IM). The national evaluation considers the calculation of the relative risk of avoidable causes of deaths (from 5 to 74 years old) due to interventions at the Brazilian Health System [25], between 2013 and 2017. This health outcome was chosen since it is a very sensitive indicator of differences in populations, enabling the identification of higher-risk groups for the implementation of special health and development programs [26].

The intra-municipal evaluation considers the calculation of the relative risk of mortality from circulatory system diseases of residents of the municipality of São Paulo, in the FU of São Paulo, by sample area, between 2006 and 2009. Death data by municipality and population by gender and age groups are publicly available by the DATASUS "Vital Statistics" and "Mortality Information System" (SIM). We obtained the São Paulo data aggregated by sample area from the "São Paulo State System for Data Analysis Foundation" (SEADE), with no access to individual information.

The motivation for choosing such outcomes and periods results from two significant theoretical concerns. The first refers to the quality of death data on the national scale. To avoid the impact of sub-notification by cause of death in the North and Northeastern regions of the country, using causes grouped in the avoidable deaths reduces uncertainty in the calculated relative risks by municipality. In this sense, using the most recent data, from 2013 to 2017, helps to reduce uncertainty since data quality has been improving year by year in Brazil. The second point concerns the intra-municipal scale. At this scale, for which death data are of high quality in the municipality of São Paulo, we chose to study deaths related to circulatory system diseases (ICD-10: I00-I99), since this is an outcome admittedly known to be associated with

socioeconomic conditions. Deaths occurred from 2006 to 2009, very close to the 2010 Demographic Census. Data were indirectly standardized by gender and detailed age range in the two scales using the software SaTScan [27].

We used simple linear regression models to validate the explanatory power of GeoSES on health. We compared the regression of outcomes with GeoSES (and its dimensions), against the regression of outcomes with HDI-M (and its components). See S1 and S2 Datasets.

It started with the verification of the assumptions for the Ordinary Least Squares regression between the outcomes and the studied indices and the spatial dependence analysis on the residues [28]. We used the geographic coordinates of the municipal headquarters in the analyzes for Brazil and the displaced coordinates of the sample areas for the intra-municipal analyzes of São Paulo. In this case, the displacement was made due to the heterogeneity of the population distribution in the peripheral sample areas, where there are dams and environmental protection sparsely populated areas. Due to the occurrence of spatial dependence on residues at both aggregation scales, we applied geographically weighted regression (GWR) models calculated in the ArcGIS 10.1 program ("Adaptive" Kernel analysis; Bandwidth Parameter method, with 53 neighbors on the national scale, and 30 in the intra-municipal). GWR is a regression that allows exploring spatial heterogeneity on data, which exists when the process being modeled varies across the study area. We evaluated the models via resulting AIC (Akaike Information Criterion) values, according to which lower values indicate a better fit of the model [29]. A spatial model with good fit should yield no spatial autocorrelation on its residues meaning that the most important variables explaining spatial variability were addressed. To verify spatial dependence on the residues, we calculated the Moran's I coefficients for the standardized residuals and their $p$ values in the GeoDa program. Moran's I coefficient measures the likelihood that an apparent spatial pattern was produced merely by chance or if there is an effect of distance on the distribution of a variable. The coefficient ranges from -1 to 1 and is equal to zero when there is no effect of the distance. We used a "Queen" neighborhood matrix, a contiguity-based relation due to the presence of irregular polygons with varying shape and surface (municipalities and sample areas). First-order Queen contiguity defines a neighbor when they have at least a point in common on their border. A significant spatial pattern was defined when p <0.05.

The results show that, on a national scale, the model with GeoSES presented a better fit (AIC: -4,583.44), when compared to the one considering the HDI-M (AIC: -524.22), although both had presented spatial dependence on its residues (Table 2). Models with the dimensions of poverty, deprivation, income, and segregation best explained the relative risk of avoidable mortality from 5 to 74 years in Brazil, without spatial dependence on their residues. Among the dimensions of this index, the model that most explains the spatial variability of risk is with the poverty dimension (Table 2). The map in Fig 3 shows the observed risks and the risks explained by GeoSES-BR/poverty. The mobility dimension was not a deterministic criterion to characterize socioeconomic differences on the national scale and did not contribute to Geo-SES-BR. Almost 44% of the spatial variability is explained by the model with the poverty dimension, which can be realized by the visual similarity among the observed and explained maps. The most striking differences among them occur in the Northern portion of the country where observed risks (map A) are higher than expected (map B) due to the socioeconomic context. This difference may be due to health assistance or another factor not addressed in the model that should be further investigated.

At the intra-municipal scale, the model with GeoSES-IM explains about 67% of the outcome variability (AIC: -357.86). In this case, the synthesis of dimensions made a definite contribution, since no single dimension outperformed GeoSES-IM (Table 3). Fig 4 allows the comparison between observed and explained values. Full regression model results are available

**Table 2. Results of simple geographically weighted regression models (GWR) between standardized relative risk of avoidable mortality from 5 to 74 years and indices (HDI-M and GeoSES-BR) and their dimensions–adjusted global R$^2$ values, Akaike Criterion Information (AIC), Moran's *I* Coefficients and *p*-value for spatial dependence on residues.**

| Indicator | Adjusted global R$^2$ | AIC | Moran's *I* Coefficient | *p*-value |
|---|---|---|---|---|
| HDI-M | 0.504 | -524.22 | 0.016 | 0.014^ |
| HDI/education | 0.406 | -3,817.45 | 0.017 | 0.008^ |
| HDI/longevity | ~ | ~ | ~ | ~ |
| HDI/income | 0.529 | -1,017.92 | 0.017 | 0.010^ |
| GeoSES-BR | 0.422 | -4,583.44 | 0.015 | 0.029^ |
| GeoSES/income | 0.428 | -4,636.58 | 0.009 | 0.107 |
| GeoSES/education | 0.434 | -3,290.28 | 0.015 | 0.013^ |
| GeoSES/wealth | 0.398 | -4,344.18 | 0.022 | 0.002^ |
| GeoSES/deprivation | 0.428 | -4,648.99 | 0.000 | 0.465 |
| GeoSES/segregation | 0.380 | -4,073.14 | 0.011 | 0.066 |
| GeoSES/poverty | 0.436 | -4,702.75* | 0.012 | 0.067 |

* best fit

^spatial dependency on residues

~local multicollinearity does not allow modeling

as S1 and S2 Tables. In the municipality of São Paulo, the model with GeoSES-IM overestimated the relative risks in the outskirts of the study area. This means that given the socioeconomic context of the areas, the observed relative risk should have been higher. Those areas correspond to parks of natural vegetation with very small populations. Thus, differences may be attributed to the effect of small population size because rates based on a few deaths are highly variable.

Therefore, GeoSES showed significant explanatory potential in both studied scales. Because health has multiple causes, including biological, behavioral, and contextual features, GeoSES is not expected to clarify all spatial variability of an outcome. Even in the area-based level, other aspects contribute to the understanding of phenomena, such as conditions of the natural and built environment, access and quality of health services, and political and macroeconomic issues.

## Discussion

Regarding the relationship between socioeconomic conditions and health, despite being a theme of the origin of Social Epidemiology, there are still disagreements as to the definition of its indicators and the contribution they exert [30–32].

The current consensus reaffirms the complexity of the theme by noting that there is no universal indicator that can explain all outcomes [9]. Thus, it is common to choose a one-dimensional variable to represent it, resulting in simplifications that do not contribute to the understanding of the studied problem. It is also trivial to use indices not designed for that purpose and that do not allow pointing the most urgent actions that should be taken by the decision-makers.

In this study, we presented an index capable of synthesizing seven dimensions that make up the socioeconomic context, allowing a global assessment of the context and its particularities from different aspects. The GeoSES does not intend to explain all the determinants of health but allows us to identify if the socioeconomic context is related to the studied outcome, how much and which aspects stand out. Residential segregation in Brazil–although conceptually very relevant–has been poorly evaluated in health studies [32]. Primarily on an intra-municipal

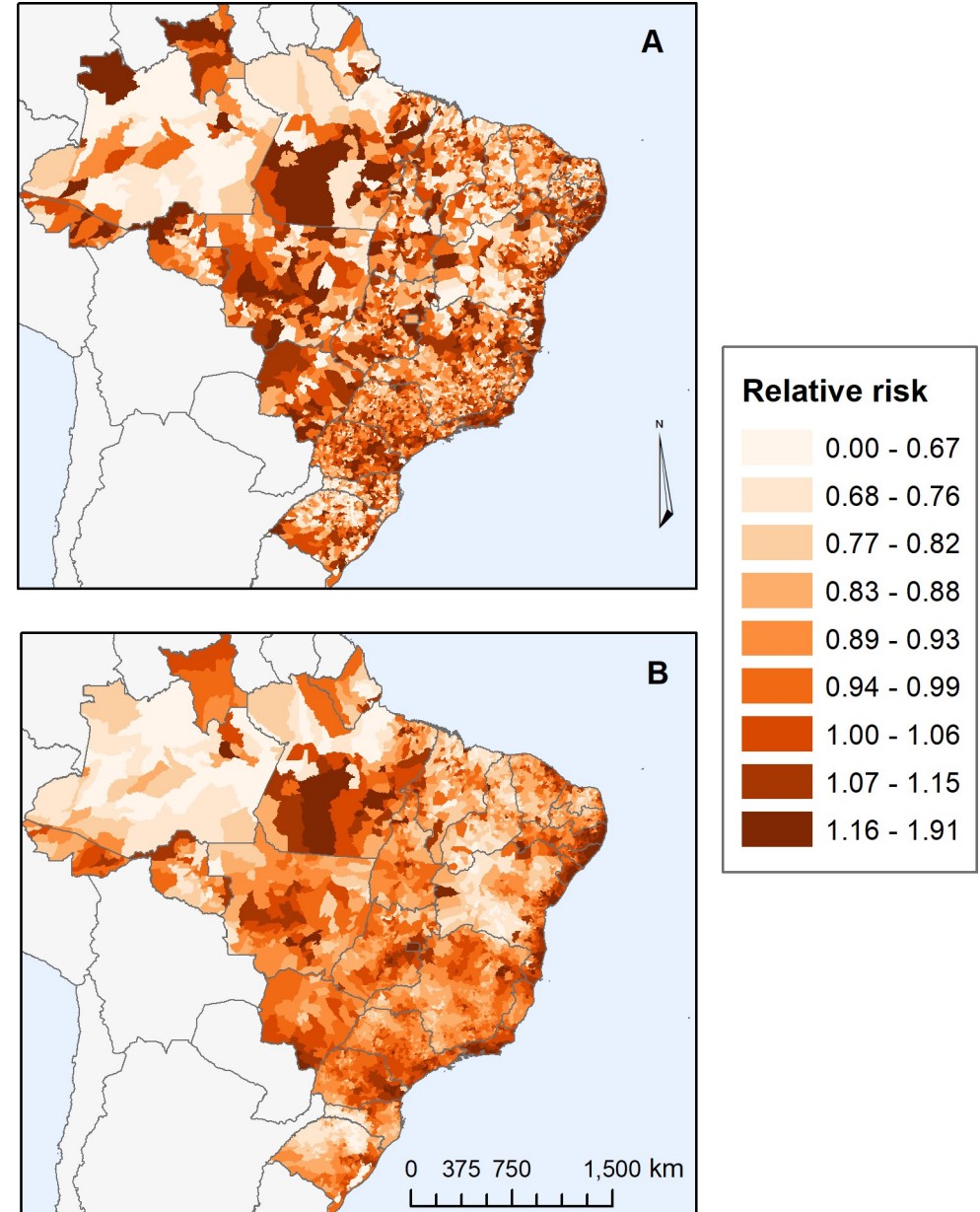

**Fig 3. Relative risks of mortality from avoidable causes of deaths (from 5 to 74 years old) due to interventions at the Brazilian health system in Brazil (2013 to 2017).** 3A) Observed relative risk. 1B) Estimated relative risk explained by the model with GeoSES-BR/poverty. Data sources: Brazilian Institute of Geography and Statistics and, Department of the Unified Health System (DATASUS). Geographic Coordinate Systems SIRGAS 2000.

scale, residential segregation is reflected in many outcomes. Highlighting it can encourage affirmative social inclusion policies. If poverty best explains the relative risk of avoidable mortality from 5 to 74 years due to interventions at the Brazilian Health System in Brazil on the national scale, it means that the focus still should be on ensuring minimum income for a part of the population.

Since we implemented GeoSES in a programming language, one can easily update it for each edition of the Demographic Census. It can also be adapted to the previous Census.

**Table 3. Results of simple linear geographically weighted regression models (GWR) between relative risks of mortality from circulatory system diseases in the municipality of São Paulo and GeoSES-IM index and its dimensions–values of adjusted global R² , Akaike Information Criterion (AIC), Moran's *I* coefficient and *p*-value for spatial dependency on residues.**

| Indicator | Adjusted global R² | AIC | Moran's *I* Coefficient | *p*-value |
|---|---|---|---|---|
| GeoSES-IM | 0.673 | -357.86* | -0.032 | 0.196 |
| GeoSES/income | 0.644 | -333.42 | -0.020 | 0.331 |
| GeoSES/education | 0.649 | -338.72 | -0.029 | 0.213 |
| GeoSES/wealth | 0.618 | -313.86 | -0.012 | 0.436 |
| GeoSES/deprivation | 0.594 | -297.72 | -0.037 | 0.146 |
| GeoSES/segregation | 0.651 | -338.15 | -0.023 | 0.298 |
| GeoSES/poverty | 0.628 | -313.28 | -0.041 | 0.117 |
| GeoSES/mobility | 0.574 | -276.01 | -0.026 | 0.261 |

* best fit

Besides, other versions of GeoSES can be developed, allowing for their enhancement by including dimensions not covered in this first version.

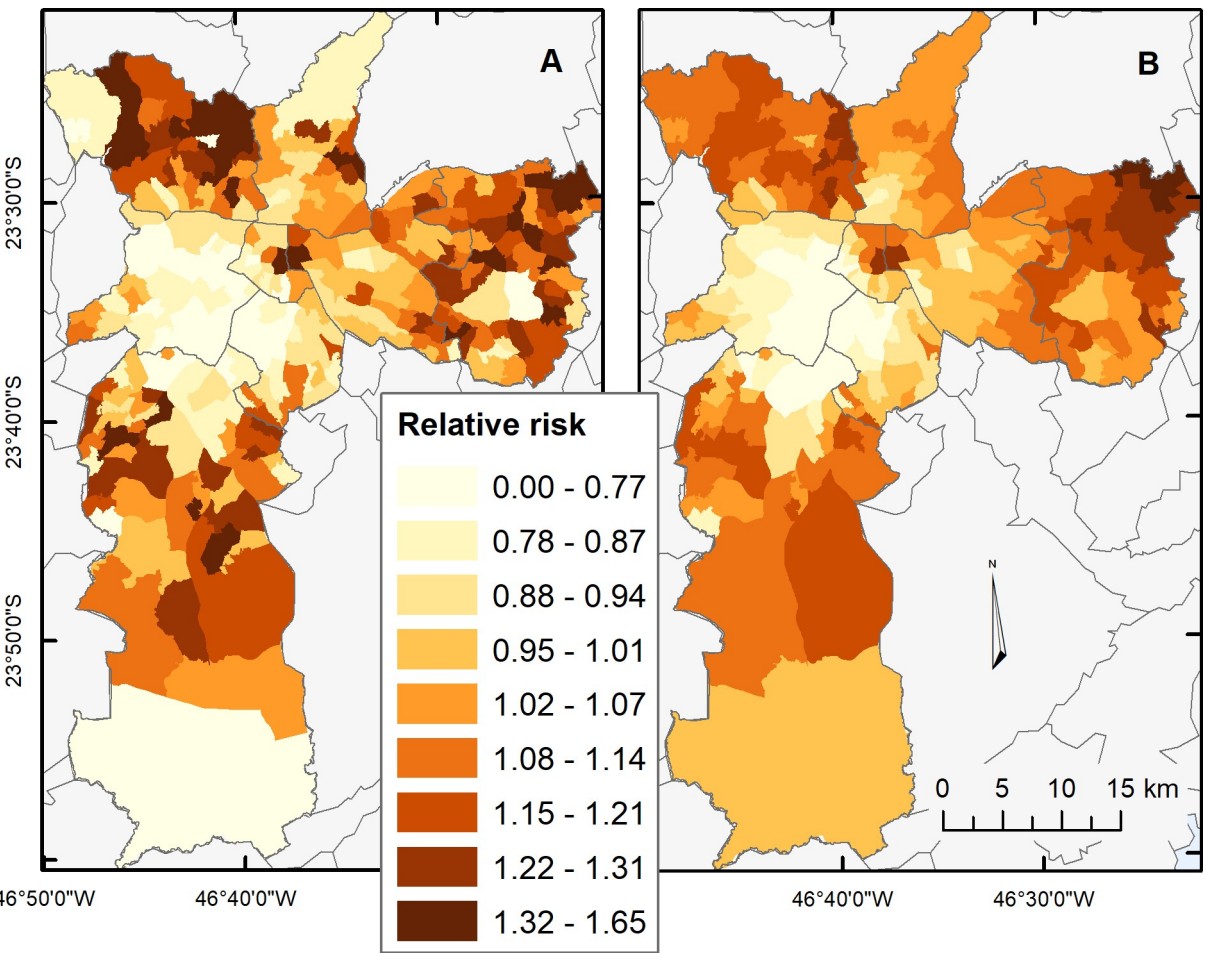

**Fig 4. Relative risks of mortality from circulatory system diseases (2006 to 2009) in São Paulo (SP).** 4A) Observed relative risk. 4B) Estimated relative risk explained by the model with GeoSES-IM. Data sources: Brazilian Institute of Geography and Statistics and, Department of the Unified Health System (DATASUS). Geographic Coordinate Systems SIRGAS 2000, UTM Projection, Fuse 23 South.

It is noteworthy, however, that there is a significant correlation of the new index with the HDI-M because the data set used to compose them is similar. Though, there is an advantage since it is possible to break the new index down to seven dimensions, unlike the HDI-M, which uses only three. Furthermore, the intra-urban index generation allows health managers to make decisions at the municipal level by observing similarities/differences among regions of the same city.

The elaboration of GeoSES starts from theoretical references according to data availability but allows the most explanatory variables to be chosen mathematically by statistical analysis. In this sense, the most explanatory variable is not arbitrary, but the most discriminating, and may differ according to the region under analysis (national, state, and municipal). It brings an innovative and insightful perspective for understanding what is most relevant in the socioeconomic context of each Brazilian region. For example, the most significant dimension at the national level is poverty; for São Paulo state, it is the residential segregation of the educational context; for the state of Rio Grande do Norte, income is the most critical dimension. Taking the sample area as an example, when comparing the principal variable in the education dimension, in the municipality of São Paulo, the percentage of people with superior level is the variable for discrimination among areas while in the municipality of Salvador (Bahia) is the percentage of people in the low educational level. Segregation seems to be central in the intra-municipal settings of the metropolitan areas, while mobility may be more relevant in other-sized municipalities. Thus, GeoSES is a robust tool for carrying studies on health and in other areas of knowledge, such as geography, sociology, and economics.

The potential contribution of a tool must be assessed given the limitations involved in its design and application. Some are inherent to the option for constructing a composite indicator. In this regard, we tried to minimize the main points when possible. One of the main objections is the selection of indicators. Here, we performed a literature search to identify the most used socioeconomic indicators on health research to include the corresponding dimensions in the composite indicator.

Another point concerns the interpretation of the index. As we defined a scale of negative and positive values, the composite index is intuitive by itself. If further information is needed to understand the variables which lead to a low GeoSES it is possible to identify the variables that compose the index in each dimension. Comparing the variables in their original units among the geographic units is a way to allow better interpretation. For instance, in the dimension "poverty" (variable P_POBREZA), it is possible to go from the best value in Brazil (2.1% in Carlos Barbosa) to the worst (91.5% in Marajá do Sena). Poverty was the most important dimension in the principal component. Policymakers should continue to focus on cash transfer programs associated with other ways to reduce it in the country.

The subject judgment also raises as a potential constraint in composite indicators. In this sense, the PCA minimizes the subjectivity since it drives the choice of the most discriminant variables, defining their weights.

Concerning the limitations in the index application, as the Brazilian demographic censuses provide data at one point in time every 10 years, the effectiveness of using Census data depends on the health outcome studied. Associations between distal social determinants and population health depend on the time lags between exposure to these risk factors and their effects on different types of health outcomes [33]. Thus, lags are widely variable. For instance, for acute or infant mortality, it makes sense to have proximity between both health and socioeconomic population data. For chronic diseases, on the other hand, lags of 20 years are expected to be more plausible [34]. Although this lag time imprecision, Census data allow a comparative spatial evaluation between the rates and risk factors and identify associations due to social-spatial inequalities occurring at the same point in time.

Other sensitive points regarding Census data are changes in the questionnaires from one Census to other and trends to cut or severely hamper Census and public health information that has occurred internationally [35]. Thus, when one chooses several variables whose meaning expresses the intended "dimension" and lets the principal component analysis point out which is the most important to explain it, the "dimension" in the two or more censuses can be compared. Actually, this is an advantage of the index, since the theoretical background of the dimensions is preserved even if there are changes in the variables among censuses. Regarding cuts in the next Census, the experimental questionnaires last available (June 2019) show a significant reduction in questions about household features and its surroundings. Due to the coronavirus pandemics in course, the Brazilian government has just decided to postpone the 2020 Demographic Census to 2021 (experimental questionnaires in the 2021 Census are available at https://www.ibge.gov.br/media/com_ mediaibge/arquivos/19361f45cc3e3b003f0a552ecde1c45f.pdf and https://www.ibge.gov.br/media/ com_mediaibge/arquivos/ee88a6181125873a8acd7b8c7ab9ad3c.pdf, in Portuguese). Comparing to the variables used in GeoSES 2010, the only affected dimension will be the material deprivation, since the ownership of various items will not be included in the basic or sample questionnaires. Based on the preliminary questionnaires, the following items will not be present in 2021: access to electric energy, ownership of television, refrigerator, cell phone, computer, motorcycle and, automobile. Nevertheless, among those that will remain, PCA will help to define the most discriminant that will help to measure material deprivation and access to public services.

The provided results–interactive maps and tabulated values by region of interest–may contribute to relevant future actions. In the scientific field, the index can support studies of the specific aspects of health inequalities and the mechanisms that lead to them. In practical terms, the index may guide the elaboration of intersectoral public policies or in state and municipal administrations.

## Supporting information

**S1 File. Abstract in portuguese.**
(PDF)

**S1 Appendix. List of input variables for creating GeoSES.**
(DOCX)

**S1 Dataset. Dataset including the variables used in the geographically weighted regressions for Brazil.**
(XLS)

**S2 Dataset. Dataset including the variables used in the geographically weighted regressions for São Paulo.**
(XLS)

**S1 Table. Geographically weighted regression results of models for the relative risk of causes of deaths (from 5 to 74 years old) in Brazil due to interventions at the Brazilian health system in Brazil (2013 to 2017).**
(DOCX)

**S2 Table. Geographically weighted regression results of models for the relative risk of mortality from circulatory system diseases in the municipality of São Paulo (2006 to 2009).**
(DOCX)

## Acknowledgments

To Armando Akira Santos Yamada, Hospital Israelita Albert Einstein, Department of Big Data Analytics, for his technical support.

## Author Contributions

**Conceptualization:** Ligia Vizeu Barrozo, Carmen Diva Saldiva de André, João Ricardo Sato, Edson Amaro Júnior.

**Data curation:** Michel Fornaciali, Guilherme Augusto Zimeo Morais.

**Formal analysis:** Ligia Vizeu Barrozo, Michel Fornaciali, Carmen Diva Saldiva de André, Guilherme Augusto Zimeo Morais, Giselle Mansur, William Cabral-Miranda.

**Funding acquisition:** Ligia Vizeu Barrozo, Carmen Diva Saldiva de André, Edson Amaro Júnior.

**Investigation:** Ligia Vizeu Barrozo, Michel Fornaciali, Carmen Diva Saldiva de André, Guilherme Augusto Zimeo Morais, Giselle Mansur, William Cabral-Miranda, Marina Jorge de Miranda.

**Methodology:** Ligia Vizeu Barrozo, Carmen Diva Saldiva de André, João Ricardo Sato.

**Project administration:** João Ricardo Sato, Edson Amaro Júnior.

**Resources:** Ligia Vizeu Barrozo, Edson Amaro Júnior.

**Software:** Michel Fornaciali, Guilherme Augusto Zimeo Morais, João Ricardo Sato.

**Supervision:** Ligia Vizeu Barrozo, João Ricardo Sato, Edson Amaro Júnior.

**Validation:** Ligia Vizeu Barrozo, Michel Fornaciali, Carmen Diva Saldiva de André, Guilherme Augusto Zimeo Morais, João Ricardo Sato.

**Visualization:** Ligia Vizeu Barrozo, Michel Fornaciali, Guilherme Augusto Zimeo Morais, Giselle Mansur, William Cabral-Miranda, Marina Jorge de Miranda.

**Writing – original draft:** Ligia Vizeu Barrozo, Michel Fornaciali.

**Writing – review & editing:** Ligia Vizeu Barrozo, Michel Fornaciali, Carmen Diva Saldiva de André, Guilherme Augusto Zimeo Morais, Giselle Mansur, William Cabral-Miranda, Marina Jorge de Miranda, João Ricardo Sato, Edson Amaro Júnior.

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
