## [Decision Letter · Decision Letter 0]

10 Feb 2020

PONE-D-19-35982

GeoSES: a socioeconomic index for health and social research in Brazil

PLOS ONE

Dear Prof Barrozo,

Thank you for submitting your manuscript to PLOS ONE. After careful consideration, we feel that it has merit but does not fully meet PLOS ONE’s publication criteria as it currently stands. Therefore, we invite you to submit a revised version of the manuscript that addresses the points raised during the review process.

This is a very interesting paper and an important contribution. Based on the reviewers´ comments and my own reading I still believe the paper needs some revision. The reviewers comments are detailed below. Please, considered the very careful suggestions they made:

revised the organization and presentation of the material review the paper to present a clear and transparent as regards the explanation and justification of the methodological choicesimprove the discussion of limitations of your approachimprove discussion (theoretical and methodological) of variables choice. 

We would appreciate receiving your revised manuscript by Mar 23 2020 11:59PM. To enhance the reproducibility of your results, we recommend that if applicable you deposit your laboratory protocols in protocols.io, where a protocol can be assigned its own identifier (DOI) such that it can be cited independently in the future. For instructions see: http://journals.plos.org/plosone/s/submission-guidelines#loc-laboratory-protocols

We look forward to receiving your revised manuscript.

Kind regards,

Bernardo Lanza Queiroz, Ph.D

Academic Editor

PLOS ONE

Journal Requirements:

2. Please include a copy of Table 3 which you refer to in your text on page 17.

3. We note you have included a table to which you do not refer in the text of your manuscript. Please ensure that you refer to Table 4 in your text; if accepted, production will need this reference to link the reader to the Table.

Reviewers' comments:

Reviewer's Responses to Questions

**Comments to the Author**

1. Is the manuscript technically sound, and do the data support the conclusions?

Reviewer #1: Yes

Reviewer #2: Partly

2. Has the statistical analysis been performed appropriately and rigorously? 

Reviewer #1: Yes

Reviewer #2: No

3. Have the authors made all data underlying the findings in their manuscript fully available?

Reviewer #1: Yes

Reviewer #2: Yes

4. Is the manuscript presented in an intelligible fashion and written in standard English?

Reviewer #1: Yes

Reviewer #2: No

5. Review Comments to the Author

Reviewer #1: This paper aims to introduce the GeoSES, a composite index that summarizes the main dimensions of the Brazilian socioeconomic context for health research purposes, to evaluate and monitor inequalities, and to develop resource and service allocation strategies.

I would first congratulate the Authors of this manuscript of their extensive work. However, there are some important issues, listed below, that should obligatory be discussed / corrected / improved.

1 - In my opinion, the main contribution of the paper is methodological and the authors should highlight and discuss this. If the authors don’t agree with my opinion, then it would be good for them to advance a little bit.

2 –Intra-municipal index (lines 102-103). I suggest that the authors bring (and discuss, if necessary) how many municipalities, among 5,565, fit this criterion. (“GeoSES-IM, for municipalities with three or more Census sample areas”).

3 - How effective is the index for health research taking into account that the data is ~10 years ago? The authors could discuss this in the discussion section, conclusion section or in a section about the limitations of the work.

4 – Lines 417-420. The authors state that the index "can easily update it for each edition of the Demographic Census." There is an important discussion in Brazil about what the 2020 Census will look like and what issues will remain in the questionnaire, since the government has already signaled for the budget cut, which will affect the size of the questionnaire and so on. Therefore, it can be imagined that some variables used in this study will no longer be there as of the next Census. How do the authors see this and what could be proposed? Furthermore, and assuming that some variables will not be in the next Census, or were not in previous censuses, how can the index be compared in time and space? I would appreciate a discussion about it in the paper.

Reviewer #2: This is a potentially interesting paper describing a new composite index of socio-economic well-being (the so-called GeoSES) for the more than 5500 municipalities in Brazil around 2010. Such index, which is constructed on the basis of the 2010 Census data, could be potentially used both by researchers and policy-makers. Yet, there are many aspects of the paper that should be improved.

1. In general, the paper is not very clearly written, and in some sections, it is confusing. While readers acquainted with the literature on the construction of composite indices might have an intuition of what the authors are talking about, this might not be the case for the general reader. In addition, some references look arbitrary or strange. The literature on epidemiology and health inequalities is huge, and the authors have chosen a curious selection for their reference list. Last but not least, the paper needs a thorough English revision.

2. The paper needs to be much more clear and transparent as regards the explanation and justification of the methodological choices. This applies to the entire “Methods” section (i.e. from page 5 to page 11). For instance:

- What is the meaning of the segregation component? From its definition, I understand it can take negative values. How is that interpreted?

- The authors are mixing all kinds of variables, very often going in opposite directions, so to say. For instance, higher values of the income variable are normatively desirable, whereas higher values of the poverty variable are normatively undesirable. If all the data crunching exercise has to have some meaning, it is crucial that all the variables point to the same direction (e.g. the higher the values, the better/the more desirable). It is unclear to me if this basic recoding exercise has been carried out by the authors.

- The subsection of “Data collection and processing” is particularly obscure.

- In the “Preprocessing” subsection the authors say they have added a value of 10 to their indicators to avoid problems with the zeroes, arguing that this does not affect the correlation structure of the data. This looks like an extremely arbitrary decision. Why not adding 1, or 100? What would happen to the results for these alternative choices? If the results are overly sensitive to the values of such constant, perhaps the use of PCA techniques is not the best one.

- When explaining the steps followed to construct the GeoSES index, it is unclear to me why do the authors need to perform PCA up to three times. I understand the first one (to generate aggregate indices within each of the six/seven dimensions) and the second one (to bring all dimensions together into an overall component). Why then a third round of PCA? More explanations and justifications should be provided.

- It would be helpful if the authors explained what do the extreme GeoSES values of -1 and 1 truly mean. That is: what would need to happen in a municipality to get a score of 1?

- Again, the subsections of “Interpretation” (page 10) and “GeoSES extensive creation” are unclear.

Very often, the authors take many things for granted and do not explain them (or even show a reference). Examples: Explaining what Moran’s I coefficient is and what it means, the same for the Geographically Weighted Regression models (GWR), the “Queen” neighborhood criterion to choose neighbors, and so on and so forth. Again, even if these are well-known methods for the specialist, they might be unknown to the general reader.

3. In the empirical section of the paper where the authors “prove” the validity of their measure by predicting certain health outcomes, the text lacks order and clarity. The authors add decorative maps (e.g. Figures 3 and 4) that are barely mentioned in the text, but nothing substantive comes out of them.

4. The authors try to sell the relevance of their approach by highlighting the limitations of currently existing approaches (e.g. like the municipal level HDI, or HDI-M). Yet, they should also point out the several limitations of their own approach. There is no such thing as “a perfect measure”, and all approaches have their advantages and disadvantages. For instance: (i) Composite indices have the advantage of simplifying complex information, at the cost of hiding important patterns that might exist in the data. (ii) In this line, composite indices are sometimes difficult to interpret, as they are made up by averaging all sorts of variables (e.g. a GeoSES score of, say, 0.1, might be obtained from high levels of poverty and low levels of segregation, or from very low education levels and high incomes, and so on). Thus, when policy makers have to make decisions on the basis of the GeoSES index, it is crucially important to know the values of the underlying variables. (iii) The use of PCA techniques might complicate comparisons over time. If the GeoSES index wants to be replicated with the new (2020?) Census or with previous censuses, the different components of the index will get different weights. Thus if the GeoSES index equals 0.2 in year 2010 and 0.3 in year 2020, we do not know if there has been a real improvement in the underlying variables or simply a variable reweighting through the PCA algorithm (or both of them simultaneously). (iv) PCA is one among several other dimensional-reduction techniques. Why not using, say, Factor Analysis?

5. In several parts of the paper, the authors state that the choice of variables included in the GeoSES index was driven by theoretical considerations. Looking at the list of selected variables and the standard questionnaires included in Censuses, I would rather say that the choice was driven by data availability issues.

6. PLOS authors have the option to publish the peer review history of their article (what does this mean?). If published, this will include your full peer review and any attached files.

Reviewer #1: No

Reviewer #2: Yes: Iñaki Permanyer

---

## [Author Response · Author response to Decision Letter 0]

23 Mar 2020

Dear Academic Editor and Reviewers,

We appreciate the editors' and reviewers' work on our article. In the revised version we addressed all the reviewers' concerns, with a significant rewriting of the main sections to present our contributions, better explaining the theoretical and methodological foundations, and also improving discussions towards the limitations of our approach as well as the choice of the variables. 

In this letter, we have addressed point by point the issues raised by the reviewers. In the manuscript itself, we have highlighted the text changes in blue, except for minor changes, to lessen clutter. 

Summary of changes 

We start with a summary of the main changes:

- Introduction: improved description of the main result and contributions; 

- Methods: 

o better explaining the segregation component; 

o including justifications for theoretical and methodological choices; 

o clarifying the algorithm steps and their objectives; 

o pointing out how to interpret the results; 

- Results: explanations of the statistical methods used for GeoSES validation;

- Discussion: extension of the applicability and limitations of our work, and the availability of the used data. 

Next, we comment the Journal Requirements and address in detail the reviewers' comments.

Journal Requirements

We checked the style requirements according to the provided links. We are confident that the final manuscript fulfills the requirements.

2. Please include a copy of Table 3 which you refer to in your text on page 17.

Thank you for the observation. Tables 3 and 4 are the same: we have numbered the tables incorrectly. Now, we corrected the number of the table in the text and in the material support files.

3. We note you have included a table to which you do not refer to the text of your manuscript. Please ensure that you refer to Table 4 in your text; if accepted, production will need this reference to link the reader to the Table.

As mentioned above, it was a numbering error: there are only three tables in this paper. Table 4 in the text and Table 3 refer to the same information.

Answers to Reviewer #1

Reviewer #1: This paper aims to introduce the GeoSES, a composite index that summarizes the main dimensions of the Brazilian socioeconomic context for health research purposes, to evaluate and monitor inequalities, and to develop resource and service allocation strategies.

I would first congratulate the Authors of this manuscript of their extensive work. However, there are some important issues, listed below, that should obligatory be discussed / corrected / improved.

We appreciate the reviewer's positive reception of our work and their comments that allow us to improve our article.

1 - In my opinion, the main contribution of the paper is methodological and the authors should highlight and discuss this. If the authors don’t agree with my opinion, then it would be good for them to advance a little bit.

We thank the reviewer for this observation, but we kindly disagree. The primarily intended contribution is to provide a tool - the index - to enhance health and social research, which need to understand/estimate the burden of the problems attributable to the different social dimensions. The contribution of the article is more in the application domain than in the methodological domain. Although the methodological aspect is crucial to the acceptance and broader utilization of the index, this approach does not represent an innovation per se, since it relies on traditional statistical analysis (mainly, Principal Component Analysis). In the application perspective, however, the use of techniques in the scales studied and in the territorial scope is unprecedented. We showed the validity of the index, and it has already been successfully used in Takano et al. (2019) and Santos et al. (2020). We highlighted our contribution in the Introduction of the final manuscript. 

2 –Intra-municipal index (lines 102-103). I suggest that the authors bring (and discuss, if necessary) how many municipalities, among 5,565, fit this criterion. (“GeoSES-IM, for municipalities with three or more Census sample areas”).

It is a good observation. The Brazilian population is spatially dispersed. In 2010, from 5,565 Brazilian municipalities, 4,967 (89,3%) had less than 50,000 inhabitants. Because of this, only 140 municipalities present three or more sample areas (that is, had more than 190,000 inhabitants, the minimum required population to define sample areas by the Census).

However, GeoSES covers the entire Brazilian population due to the BR and UF levels. We can assess almost 45% of the population at an even more detailed level, through the IM level, which is a novelty. Although there are few municipalities with 190k inhabitants, we are talking about the most significant areas in terms of population density.

3 - How effective is the index for health research taking into account that the data is ~10 years ago? The authors could discuss this in the discussion section, conclusion section or in a section about the limitations of the work.

Associations between distal social determinants and population health depend on the time lags between exposure to these risk factors and their effects on different types of health outcomes (LYNCH et al., 2005). Thus, lags are widely variable. For instance, for acute or infant mortality, it makes sense to have proximity between both health and population data. For chronic diseases related to the consumption of alcohol and tobacco, on the other hand, lags of 20 years are expected to be more plausible (JIANG et al., 2018). In the literature, most studies try to keep health data close to the Census to have a better denominator of the population for the calculated rates. Few studies have discussed this critical issue. Kim (2019) used average lag periods between 3 to 17 years, according to the different social determinants analyzed to study firearm-related homicides. Studying life expectancy, OECD (2017) used explanatory factors lagged by five years to account for the delayed effects on health. The association between Social Determinants of Health indices and premature mortality (defined as death before age 75 years) in Chicago was measured by years of potential life lost and aggregated to a 5-year mean (2009-2013) and calculated as an age-adjusted rate at the census tract level. All variables derive from the 2014 American Community Survey 5-year mean (KOLAK et al., 2020). Thus, the length of the lag is not precise in the literature yet. Although this lag time imprecision, Census data allow a comparative spatial evaluation between the rates and risk factors and to identify associations due to social-spatial inequalities occurring at the same point in time. We introduced such a discussion and references to the final manuscript. 

4 – Lines 417-420. The authors state that the index "can easily update it for each edition of the Demographic Census." There is an important discussion in Brazil about what the 2020 Census will look like and what issues will remain in the questionnaire, since the government has already signaled for the budget cut, which will affect the size of the questionnaire and so on. Therefore, it can be imagined that some variables used in this study will no longer be there as of the next Census. How do the authors see this and what could be proposed? Furthermore, and assuming that some variables will not be in the next Census, or were not in previous censuses, how can the index be compared in time and space? I would appreciate a discussion about it in the paper.

We appreciate the reviewer for bringing this point to the light. It surely deserves a more in-depth discussion. We start it quoting Wilson et al., (2017), stating that “over the past few years, trends to cut or severely hamper census and public health information have occurred internationally”. Such a statement relates to Canada, the United Kingdom, the United States of America, and the European Union. 

Brazil follows the same lack of cost-benefit analysis of the Census related to public policies and public health interventions. Due to the coronavirus pandemics in course, the Brazilian government has just decided to postpone the 2020 Demographic Census to 2021. Regarding cuts in the next Brazilian Demographic Census, the experimental questionnaires last available (June 2019) show a significant reduction in questions about household features and its surroundings (experimental questionnaires in the 2020 Census are available at https://www.ibge.gov.br/media/com_mediaibge/arquivos/19361f45cc3e3b003f0a552ecde1c45f.pdf and https://www.ibge.gov.br/media/com_mediaibge/arquivos/ee88a6181125873a8acd7b8c7ab9ad3c.pdf, in Portuguese).

Brazilian censuses always have suffered changes from one decade to others, for other reasons beyond budget cuts. For instance, when we intend to express the material deprivation dimension measuring how many material resources and conveniences that are part of modern life a person has (such as adequate housing, car ownership, refrigerator, computer, among others) the questionnaire must include goods that are available for a significant part of the population. In 2000, the cell phone was not a good widely afforded for Brazilians and did not enter in the questionnaire. However, in 2010 cell phone was added to the questionnaire, and other items as video cassette recorder and microwave oven were no longer inventoried as they had been in 2000. Thus, when one chooses several variables whose meaning expresses the intended “dimension” and lets the principal component analysis (PCA) point out which one best explains the variability of the data, the "dimension" in the two or more censuses can be compared.That is an advantage of the index since the theoretical background of the dimensions is preserved even if there are changes in the variables among censuses.

Comparing to the variables used in GeoSES 2010, the only affected dimension will be the material deprivation, since the ownership of various items will not be included in the basic or sample questionnaires. The following items will not be present in 2020: access to electric energy, ownership of television, refrigerator, cell phone, computer, motorcycle and, automobile. Nevertheless, among those that will remain, PCA will help to select the variable that best explains the material deprivation and access to public services. 

We incorporated such a discussion in the final manuscript. 

Answers to Reviewer #2

Reviewer #2: This is a potentially interesting paper describing a new composite index of socio-economic well-being (the so-called GeoSES) for the more than 5500 municipalities in Brazil around 2010. Such index, which is constructed on the basis of the 2010 Census data, could be potentially used both by researchers and policy-makers. Yet, there are many aspects of the paper that should be improved.

We thank the reviewer for recognizing the academic and government potential of our work.

1. In general, the paper is not very clearly written, and in some sections, it is confusing. While readers acquainted with the literature on the construction of composite indices might have an intuition of what the authors are talking about, this might not be the case for the general reader. In addition, some references look arbitrary or strange. The literature on epidemiology and health inequalities is huge, and the authors have chosen a curious selection for their reference list. Last but not least, the paper needs a thorough English revision.

We acknowledge that some used terms are specific-area related. We respond to this suggestion by broadening the understanding of terms for general readers. We also made another English revision to enhance the writing.

Regarding literature choice, we honestly and kindly disagree with the reviewer. We cited papers that addressed some issues related to the individual x area-level socioeconomic indicators, the choice of socioeconomic indicators for ecologic studies, methods to define a socioeconomic index, or related to the social determinants of health. Some important specific issues are sometimes described as not the main topic of the cited papers and published in journals not dedicated to Epidemiology, which leads to the impression that the references are curious. Nevertheless, all of them support the ideas where we cited them in the text. Michel Marmot, Nancy Krieger, and Anne V. Diez-Roux, for instance, are relevant authors from the Epidemiology and Social Epidemiology literature.

2. The paper needs to be much more clear and transparent as regards the explanation and justification of the methodological choices. This applies to the entire “Methods” section (i.e. from page 5 to page 11). For instance:

- What is the meaning of the segregation component? From its definition, I understand it can take negative values. How is that interpreted?

We better explained the segregation component, indicating that it varies from -1 (most deprived) to 1 (most privileged). A negative ICE means that the area presents more people in the condition of deprivation than in the higher extreme; a positive value, the opposite. A value of zero indicates that extreme concentrations of either of the two groups do not dominate the area.

- The authors are mixing all kinds of variables, very often going in opposite directions, so to say. For instance, higher values of the income variable are normatively desirable, whereas higher values of the poverty variable are normatively undesirable. If all the data crunching exercise has to have some meaning, it is crucial that all the variables point to the same direction (e.g. the higher the values, the better/the more desirable). It is unclear to me if this basic recoding exercise has been carried out by the authors.

It is not crucial that all the variables point to the same direction. The principal components analysis is concerned with explaining the variance-covariance structure through few linear combinations of the original variables (JOHNSON; WICHERN, 2007, Chapter 8). Variables that point to opposite directions have negative covariance and correlation, and their coefficients in the linear combinations (principal components) will have opposite signs. Examples can be found in Johnson and Whichern (2007) and Lalloué et al. (2013).

- The subsection of “Data collection and processing” is particularly obscure.

This section has two main objectives: a) to show how IBGE organize and made available the original data, and b) to describe how we treat such data to express its information in terms of percentage coverage of the population. For example, if there was initially a count of people without instruction in a municipality, we are now interested in knowing what percentage of the municipality's population this count represents.

We believe that objective b) is clear and justified in this and other parts of the text. However, objective a) may not be visible to readers who are not familiar with how IBGE makes Census data available. We thank the reviewer for calling our attention to this point, so we edited the text to characterize both objectives of the section better.

- In the “Preprocessing” subsection the authors say they have added a value of 10 to their indicators to avoid problems with the zeroes, arguing that this does not affect the correlation structure of the data. This looks like an extremely arbitrary decision. Why not adding 1, or 100? What would happen to the results for these alternative choices? If the results are overly sensitive to the values of such constant, perhaps the use of PCA techniques is not the best one.

The PCA is sensitive to data when identifying correlations between variables. However, the constant value was added to all data in the sample, causing its statistical distribution to remain the same. The purpose of this sum is to avoid the instability of the method during matrix inversion. Indeed, the choice of value 10 is random, but any constant value would have the same effect.

- When explaining the steps followed to construct the GeoSES index, it is unclear to me why do the authors need to perform PCA up to three times. I understand the first one (to generate aggregate indices within each of the six/seven dimensions) and the second one (to bring all dimensions together into an overall component). Why then a third round of PCA? More explanations and justifications should be provided.

We thank the reviewer for calling our attention to this topic. In the final manuscript, we better explained the purpose of each step: 

1) The objective in this first step was to generate aggregate indices within each of the dimensions, and a PCA was performed in each of them. The number of components selected was such that the percentage of total variance explained was greater than or equal to 75%. For ease of interpretation, we considered the variables with the highest coefficient in each component.

2) Considering the variables selected in step 1, we applied another PCA, and we considered its first principal component. The objective here was to bring all dimensions together into an overall component.

3) In order to eliminate variables that have little contribution to the index obtained in step 2, only those whose absolute coefficient values were below the average of the coefficients were eliminated, and we applied the PCA method to the remaining variables. The resulting first component defines the GeoSES (Socioeconomic Index of Geographic Context for Health and Social Studies).

Also, we added the justification for applying the third PCA step: “In fact, the index obtained in step 2 could have been considered the final one. However, for its calculation, we would need to know the values of variables that do not have an important contribution. In Step 3, these variables were eliminated, and the coefficients of the remainder were calculated again, following the procedure presented in Lalloué (2013).”

- It would be helpful if the authors explained what do the extreme GeoSES values of -1 and 1 truly mean. That is: what would need to happen in a municipality to get a score of 1?

Extreme GeoSES values mean the worst and the best socioeconomic contexts in Brazil (for the national scale), in the Federation Unit (in the state) and, in the municipality (in the intramunicipal setting). In other words, to have a GeoSES of +1 at the national level, the municipality must have the best relative socioeconomic context of the country in the most discriminating variable. 

We included the text below into the “Interpretation” section to clarify this point:

“Extreme GeoSES values mean the worst (-1) and the best (1) socioeconomic contexts in the analyzed scale. In other words, to have a GeoSES of 1 at the national level, the municipality must have the best relative socioeconomic context of the country in the most discriminating variable. So, if in 2010 a municipality had a GeoSES index of 0.2 and in 2021 the index is 0.3, there would be a relative improvement in the socioeconomic context because even with possible changes in the most discriminating variables, this municipality is closest to the best municipality than it was in 2010.”

- Again, the subsections of “Interpretation” (page 10) and “GeoSES extensive creation” are unclear.

In the section "Interpretation" we show how GeoSES can also be analyzed in terms of the dimensions that compose it, in addition to the final value itself. We can decompose GeoSES into multiple dimensions, so we use data from the most significant variable for each activated dimension to illustrate how they contribute to the final score. For example, in terms of Education, a municipality may have that dimension represented by the percentage of people with complete higher education. The perception of the "advance" or "backwardness" of this municipality in this dimension must also be carried out in comparison with other municipalities.

The section "GeoSES extensive creation" tells how the methodology developed and evaluated for the municipality of São Paulo was implemented in programming language for the generation of GeoSES throughout the national territory, at its three levels: BR, UF, and IM. The section ends by describing computational aspects of the resulting source code.

We have edited both sections to make reading and understanding easier for the general reader.

Very often, the authors take many things for granted and do not explain them (or even show a reference). Examples: Explaining what Moran’s I coefficient is and what it means, the same for the Geographically Weighted Regression models (GWR), the “Queen” neighborhood criterion to choose neighbors, and so on and so forth. Again, even if these are well-known methods for the specialist, they might be unknown to the general reader.

Indeed, the text needs clarity on these points to be more accessible to a broader public. We explained these particularities, as follows. The blue text is the new one. 

FROM: Due to the occurrence of spatial dependence on residues at both aggregation scales, we applied geographically weighted regression models calculated in the ArcGIS 10.1 program (“Adaptive” Kernel analysis; Bandwidth Parameter method, with 53 neighbors on the national scale, and 30 in the intra-municipal). We evaluated the models via resulting AIC (Akaike Information Criterion) values, according to which lower values indicate a better fit of the model [29]. To verify spatial dependence on the residues, we calculated the Moran’s I coefficients for the standardized residuals and their p values in the GeoDa program. 

TO: Due to the occurrence of spatial dependence on residues at both aggregation scales, we applied geographically weighted regression (GWR) models calculated in the ArcGIS 10.1 program (“Adaptive” Kernel analysis; Bandwidth Parameter method, with 53 neighbors on the national scale, and 30 in the intra-municipal). GWR is a regression that allows exploring spatial heterogeneity on data, which exists when the process being modeled varies across the study area. We evaluated the models via resulting AIC (Akaike Information Criterion) values, according to which lower values indicate a better fit of the model [29]. A spatial model with good fit should yield no spatial autocorrelation on its residues meaning that the most important variables explaining spatial variability were addressed. To verify spatial dependence on the residues, we calculated the Moran’s I coefficients for the standardized residuals and their p values in the GeoDa program. Moran’s I coefficient measures the likelihood that an apparent spatial pattern was produced merely by chance or if there is an effect of distance on the distribution of a variable. The coefficient ranges from -1 to 1 and is equal to zero when there is no effect of the distance. We used a “Queen” neighborhood matrix, a contiguity-based relation due to the presence of irregular polygons with varying shape and surface (municipalities and sample areas). First-order Queen contiguity defines a neighbor when they have at least a point in common on their border. A significant spatial pattern was defined when p <0.05. 

3. In the empirical section of the paper where the authors “prove” the validity of their measure by predicting certain health outcomes, the text lacks order and clarity. The authors add decorative maps (e.g. Figures 3 and 4) that are barely mentioned in the text, but nothing substantive comes out of them.

We enhanced the descriptions of Figures 3 and 4, describing how the observed results confirm the correlations of socioeconomical development with the phenomena under study. For the sake of completeness, we reproduce here the further introduced explanations:

For Figure 3: “Almost 44% of the spatial variability is explained by the model with the poverty dimension, which can be realized by the visual similarity among the observed and explained maps (Fig 3). The most striking differences among them occur in the Northern portion of the country where observed risks (map A) are higher than expected (map B) due to the socioeconomic context. This difference may be due to health assistance or another factor not addressed in the model that should be further investigated.”

For Figure 4: “In the municipality of São Paulo, GeoSES-IM overestimated the relative risks in the outskirts of the study area. This means that given the socioeconomic context of the areas, the observed relative risk should have been higher. Those areas correspond to parks of natural vegetation with very small populations. Thus, differences may be attributed to the effect of small population size because rates based on a few deaths are highly variable and may be unreliable.” 

4. The authors try to sell the relevance of their approach by highlighting the limitations of currently existing approaches (e.g. like the municipal level HDI, or HDI-M). Yet, they should also point out the several limitations of their own approach. There is no such thing as “a perfect measure”, and all approaches have their advantages and disadvantages. For instance:

(i) Composite indices have the advantage of simplifying complex information, at the cost of hiding important patterns that might exist in the data. 

Indeed, this is one of the critiques of the composite index. Because of this constraint, we believe that keeping the different dimensions with few variables in each avoid hiding the patterns. This approach allows an in-depth analysis of the social context. We addressed this point, adding the following text into the Discussion section of the final manuscript:

“The potential contribution of a tool must be assessed in view of the limitations involved in its design and application. Some are inherent to the option for constructing a composite indicator. In this regard, we tried to minimize the main points when possible. One of the main objections is the selection of indicators. Here, we performed a literature search to identify the most used socioeconomic indicators on health research to include the corresponding dimensions in the composite indicator. 

Another point concerns the interpretation of the index. As we defined a scale of negative and positive values, the composite index is intuitive by itself. If further information is needed to understand the variables which lead to a low GeoSES it is possible to identify the variables that compose the index in each dimension. Comparing the variables in their original units among the geographic areas is a way to allow better interpretation. For instance, in the dimension “poverty” (variable P_POBREZA), it is possible to go from the best value in Brazil (2.1% in Carlos Barbosa) to the worst (91.5% in Marajá do Sena). Poverty was the most important dimension in the principal component. Policymakers should focus on cash transfer programs or other ways to reduce it in the country.

The subject judgment also raises as a potential constraint in composite indicators. In this sense, the PCA minimizes the subjectivity since it drives the choice of the most important variables, defining their weights.” 

(ii) In this line, composite indices are sometimes difficult to interpret, as they are made up by averaging all sorts of variables (e.g. a GeoSES score of, say, 0.1, might be obtained from high levels of poverty and low levels of segregation, or from very low education levels and high incomes, and so on). Thus, when policy makers have to make decisions on the basis of the GeoSES index, it is crucially important to know the values of the underlying variables. 

We agree with the reviewer that "when policy makers have to make decisions on the basis of the GeoSES index, it is crucially important to know the values of the underlying variables". Multiple combinations of dimensions and variables can explain the final GeoSES score. We emphasize here, again, that GeoSES must be interpreted in a comparative way between regions (municipalities or weighted areas) of the same analysis (BR, UF, or IM). The score alone does not have an isolated semantic value, as we have pointed out in other responses and highlighted in the text of the revised manuscript. That is why we introduced in section "Interpretation" how values must be understood.

(iii) The use of PCA techniques might complicate comparisons over time. If the GeoSES index wants to be replicated with the new (2020?) Census or with previous censuses, the different components of the index will get different weights. Thus if the GeoSES index equals 0.2 in year 2010 and 0.3 in year 2020, we do not know if there has been a real improvement in the underlying variables or simply a variable reweighting through the PCA algorithm (or both of them simultaneously). 

We addressed this point in a previous question. The primary purpose of the index is to allow to discriminate the differences among the socioeconomic contexts and spatial comparison among the geographic units in the considered level of aggregation. The best socioeconomic context (+1) drives a ranking among the following decreasing values. So, if in 2010 a municipality has a GeoSES index of 0.2 and in 2020 the index is 0.3, there was a relative improvement in the socioeconomic context because even with possible changes in the most discriminating variables, this municipality is closest to the municipality with the best social context (+1) than it was in 2010.

 (iv) PCA is one among several other dimensional-reduction techniques. Why not using, say, Factor Analysis?

Indeed, we could have used a method of factor analysis. However, we would have to assume an a priori model. Furthermore, it is well known that in the Principal Components Solution of the Factor Model, which does not suppose any multivariate distribution for the data, the factor loadings are the scaled coefficients of the principal components. Thus, the results obtained by PCA and factor analysis will be, in this case, equivalents.

5. In several parts of the paper, the authors state that the choice of variables included in the GeoSES index was driven by theoretical considerations. Looking at the list of selected variables and the standard questionnaires included in Censuses, I would rather say that the choice was driven by data availability issues.

The comment is very pertinent. Besides theoretical considerations, data availability also drove the choice of the variables. What we meant is that it was not a choice based on our assumptions but that we chose the variables based on the literature from the available variables in the Brazilian Census.

Nevertheless, for the sake of completeness, we include mentions of data availability along with the text of the final manuscript.

References

Here are the references cited in this Response to Reviewers:

JIANG, H.; LIVINGSTON, M.; ROOM, R.; CHENHALL, R.; ENGLISH, D. R. Temporal Associations of Alcohol and Tobacco Consumption With Cancer Mortality. JAMA Network Open, v. 1, n. 3, p. e180713, 13 jul. 2018. Disponível em: http://jamanetworkopen.jamanetwork.com/article.aspx?doi=10.1001/jamanetworkopen.2018.0713.

JOHNSON, R. A.; WICHERN, D. W. Applied Multivariate Statistical Analysis. 6th. ed. New Jersey: Prentice Hall, 2007. 

KIM, D. Social determinants of health in relation to firearm-related homicides in the United States: A nationwide multilevel cross-sectional study. PLOS Medicine, v. 16, n. 12, p. e1002978, 17 dez. 2019. Disponível em: https://dx.plos.org/10.1371/journal.pmed.1002978.

KOLAK, M.; BHATT, J.; PARK, Y. H.; PADRÓN, N. A.; MOLEFE, A. Quantification of Neighborhood-Level Social Determinants of Health in the Continental United States. JAMA Network Open, v. 3, n. 1, p. e1919928, 29 jan. 2020. Disponível em: https://jamanetwork.com/journals/jamanetworkopen/fullarticle/2759757.

LALLOUÉ, B.; MONNEZ, J.-M.; PADILLA, C.; KIHAL, W.; LE MEUR, N.; ZMIROU-NAVIER, D.; DEGUEN, S. A statistical procedure to create a neighborhood socioeconomic index for health inequalities analysis. International Journal for Equity in Health, v. 12, n. 21, p. 1–11, 2013. 

LYNCH, J.; HARPER, S.; KAPLAN, G. A.; DAVEY SMITH, G. Associations Between Income Inequality and Mortality Among US States: The Importance of Time Period and Source of Income Data. American Journal of Public Health, v. 95, n. 8, p. 1424–1430, ago. 2005. Disponível em: http://ajph.aphapublications.org/doi/10.2105/AJPH.2004.048439.

OECD. Health at a Glance 2017. [s.l.] OECD, 2017. 

SANTOS, N. V. dos; VIEIRA, C. L. Z.; SALDIVA, P. H. N.; PACI MAZZILLI, B.; SAIKI, M.; SAUEIA, C. H.; DE ANDRÉ, C. D. S.; JUSTO, L. T.; NISTI, M. B.; KOUTRAKIS, P. Levels of Polonium-210 in brain and pulmonary tissues: Preliminary study in autopsies conducted in the city of Sao Paulo, Brazil. Scientific Reports, v. 10, n. 1, p. 180, 13 dez. 2020. Disponível em: http://www.nature.com/articles/s41598-019-56973-z.

TAKANO, A. P. C.; JUSTO, L. T.; DOS SANTOS, N. V.; MARQUEZINI, M. V.; DE ANDRÉ, P. A.; DA ROCHA, F. M. M.; PASQUALUCCI, C. A.; BARROZO, L. V.; SINGER, J. M.; DE ANDRÉ, C. D. S.; SALDIVA, P. H. N.; VERAS, M. M. Pleural anthracosis as an indicator of lifetime exposure to urban air pollution: An autopsy-based study in Sao Paulo. Environmental Research, v. 173, p. 23–32, jun. 2019. Disponível em: https://linkinghub.elsevier.com/retrieve/pii/S0013935119301343.

WILSON, R. T.; HASANALI, S. H.; SHEIKH, M.; CRAMER, S.; WEINBERG, G.; FIRTH, A.; WEISS, S. H.; SOSKOLNE, C. L. Challenges to the census: international trends and a need to consider public health benefitsPublic Health, 2017. .

---

## [Editor Report · Decision Letter 1]

8 Apr 2020

GeoSES: a socioeconomic index for health and social research in Brazil

PONE-D-19-35982R1

Dear Dr. Barrozo,

We are pleased to inform you that your manuscript has been judged scientifically suitable for publication and will be formally accepted for publication once it complies with all outstanding technical requirements.

With kind regards,

Bernardo Lanza Queiroz, Ph.D

Academic Editor

PLOS ONE
---

## [Editor Report · Acceptance letter]

17 Apr 2020

PONE-D-19-35982R1 

GeoSES: a socioeconomic index for health and social research in Brazil 

Dear Dr. Barrozo:

I am pleased to inform you that your manuscript has been deemed suitable for publication in PLOS ONE. Congratulations! Your manuscript is now with our production department. 

With kind regards,

on behalf of

Dr. Bernardo Lanza Queiroz 

Academic Editor

PLOS ONE